# Data Resource Profile: The Multiple Sclerosis Documentation System 3D and AOK PLUS Linked Database (MSDS-AOK PLUS)

**DOI:** 10.3390/jcm12041441

**Published:** 2023-02-10

**Authors:** Marco Ghiani, Evi Zhuleku, Anja Dillenseger, Ulf Maywald, Andreas Fuchs, Thomas Wilke, Tjalf Ziemssen

**Affiliations:** 1Institut für Pharmakoökonomie und Arzneimittellogistik e.V. (IPAM e.V.), 23966 Wismar, Germany; 2Cytel Inc., 10785 Berlin, Germany; 3ZKN, Zentrum für Klinische Neurowissenschaften, Neurologische Klinik und Poliklinik für Neurologie, Universitätsklinikum Carl Gustav Carus, 01307 Dresden, Germany; 4AOK PLUS, 01067 Dresden, Germany

**Keywords:** multiple sclerosis, linked database, real-world evidence, administrative claims, electronic medical records

## Abstract

Real-world evidence in multiple sclerosis (MS) is limited by the availability of data elements in individual real-world datasets. We introduce a novel, growing database which links administrative claims and medical records from an MS patient management system, allowing for the complete capture of patient profiles. Using the AOK PLUS sickness fund and the Multiple Sclerosis Documentation System MSDS^3D^ from the Center of Clinical Neuroscience (ZKN) in Germany, a linked MS-specific database was developed (MSDS-AOK PLUS). Patients treated at ZKN and insured by AOK PLUS were recruited and asked for informed consent. For linkage, insurance IDs were mapped to registry IDs. After the deletion of insurance IDs, an anonymized dataset was provided to a university-affiliate, IPAM e.V., for further research applications. The dataset combines a complete record of patient diagnoses, treatment, healthcare resource use, and costs (AOK PLUS), with detailed clinical parameters including functional performance and patient-reported outcomes (MSDS^3D^). The dataset currently captures 500 patients; however, is actively expanding. To demonstrate its potential, we present a use case describing characteristics, treatment, resource use, and costs of a patient subsample. By linking administrative claims to clinical information in medical charts, the novel MSDS-AOK PLUS database can increase the quality and scope of real-world studies in MS.

## 1. Introduction

Real-world data (RWD) are used to address an increasing number of questions related to multiple sclerosis’ (MS) course of disease, prognosis, and treatment [1]. Real-world evidence (RWE) of this type can also be important to payers and healthcare decision makers to inform pricing, reimbursement, and labels for new therapies [2]. For such studies, RWD on MS patients can be collected from a large variety of sources, including product and disease registries, electronic medical records (EMRs), and administrative claims databases [1,2,3,4]. However, each data source individually presents specific limitations that restrict the number of questions that can be addressed and hinder the accuracy of the results, thereby introducing bias [1]. For example, while claims data offer a comprehensive picture of patients’ healthcare resource use and costs, as well as comorbidities, they lack information on disease severity and duration, laboratory results, and important prognostic measures such as the Expanded Disability Status Scale (EDSS) [5,6]. On the other hand, data sources such as registries and EMRs, which are richer in terms of documenting patients’ clinical characteristics and disease severity, lack information on the economic burden of the disease and/or the broader comorbidity profile of the patient.

While linking different data sources represents a unique opportunity to bring together the strengths of each database, patient-level linking remains a challenge, and linked databases in MS are rare [7]. To date, there have been only a few studies employing linked RWD data sources in MS. Specifically, a prior study linked EMRs from 15 MS centers in France with data from the national death register to collect information on mortality [8]. Another study linked clinical and population-based health administrative databases from two Canadian provinces to investigate the association between psychiatric comorbidity and MS disability progression [9]. The NARCOMS Registry in the US conducted a feasibility study to ascertain the possibility to link registry data to health claims data. Of 1000 registry participants, they achieved linking for 698 participants [10]. Furthermore, one recent US study linked claims data for MS patients to EMRs [11]. The specific EMR source used in this study lacked complete information on the EDSS score, which they supplemented using a claims-based proxy for assessing patient disability status. Another US study linked claims and EMR data to validate a claims-based proxy to identify MS cases [12]. Moreover, a study conducted in the province of Manitoba in Canada linked the MS Clinic registry to administrative datasets to develop and validate an algorithm for disability in MS [13]. Standing databases combining the power of various RWD sources for further RWE studies and health economic and outcome research remain scarce.

To address this important data gap, the Institut für Pharmakoökonomie und Arzneimittellogistik e.V. (IPAM e.V.) and the Center of Clinical Neuroscience (ZKN), University Clinic Carl Gustav Carus at the Technical University Dresden in Germany, partnered with the German statutory health insurance, AOK PLUS, to generate a pioneering linked database, the MSDS-AOK PLUS. This dataset combines health claims with patient records from the Multiple Sclerosis Documentation System MSDS^3D^, a high-quality computer-based patient management system developer at ZKN, which allows the documentation and management of visit schedules and mandatory examinations by integration of data input from patients, attending physicians and nurses [14,15,16]. This paper aims to introduce the novel MSDS-AOK PLUS database and describe its data, features, and future developments.

## 2. Data Resource Basics

MSDS-AOK PLUS is a growing linked database collecting information from administrative claims data provided by AOK PLUS and EMRs from the MSDS^3D^ patient management system.

### 2.1. MSDS^3D^

MSDS was developed to improve the documentation and management of patients with MS, and has been made available in different versions over the years (MSDS clinic, MSDS practice, MSDS^3D^) [16,17,18,19]. The latest development, MSDS^3D^ was created at the beginning of 2010 by the MSDS project group in Dresden, Saxony [14,15]. The software allows the input of patients’ personal data, a structured history and clinical examination, clinical scores, and treatment details. The highly specialized system has been used for the Multiple Sclerosis Partners Advancing Technology and Health Solutions (MS PATHS), an initiative that integrates routine clinical care with both standardized patient reported outcomes (PROs) and standardized magnetic resonance imaging (MRI) sequences to better understand MS and subsequently approaches to personalized medicine [20]. With various therapy-specific modules, MSDS^3D^ serves as a key platform for post-marketing effectiveness, comparative, or post-authorization safety studies [15,21,22,23,24,25,26]. Specifically, MSDS^3D^ consists of data on patient MS clinical history and routine care, including treatment of MS symptoms and disease-modifying therapy (DMT) use. Moreover, the system allows for the documentation of functional performance, including cognitive testing, gait analysis, MS performance testing (MSPT), EDSS, and imaging procedures (MRI and OCT) over time [27]. Notably, MSDS^3D^ includes patient reported outcome (PROs) questionnaires, among which are the MS-Health Resource Survey (MS-HRS), Quality of Life in Neurological Disorders (NeuroQoL), Early Mobility Impairment Questionnaire (EMIQ), and the Multiple Sclerosis Walking Scale (MSWS-12) [28,29,30,31]. Currently, the connection to a patient portal is being established, which, in addition to communication and data exchange between doctor and patient, enables patients to feed in documents and information themselves.

### 2.2. AOK PLUS

Approximately 85% of the German population is insured by statutory health insurance (SHI) funds [32]. Specifically, the AOK PLUS sickness fund consists of data on 3.4 million patients insured by AOK PLUS in the regions of Saxony and Thuringia. The dataset provides nearly a full coverage of healthcare-related services, including a record of all inpatient and outpatient diagnoses and procedures, inpatient hospital admissions, drug use, as well as outpatient general practitioner (GP) and specialist visits. In addition to healthcare resource utilization, the sickness fund allows for the determination of patient costs in various settings, including primary care, hospitalizations, rehabilitation, prescriptions, and indirect costs from inability to work. In healthcare research, the dataset is used for a multitude of studies, including but not limited to epidemiological assessments, description of treatment patterns, investigating disease burden, comparative effectiveness, and evaluation of healthcare resource use and costs.

### 2.3. Linkage: MSDS-AOK PLUS

To generate the linked database, patients within the MSDS^3D^ system insured by AOK PLUS were recruited during regular visits to ZKN and asked to provide informed consent (IC) for their inclusion. ZKN then transferred a list of pseudonymized registry IDs and AOK PLUS insurance numbers pertaining to the patient to AOK PLUS for linkage (Figure 1). Using the insurance number, the registry ID was mapped to a pseudonymized claims data ID. After subsequent deletion of patient insurance numbers, a linked complementary dataset was generated and is currently accessible for subsequent analysis through IPAM e.V., a university-affiliated research institute. As of February 2023, the database consisted of 500 patients; however, recruitment and subsequent linkage is ongoing and expected to increase at a rate of 200–250 patients per year. A total of approximately 900 AOK PLUS insured patients are currently included in the MSDS^3D^ system. As a future directive, patients insured by AOK PLUS and treated at other neurological practices which employ MSDS^3D^ can also be included in the linked dataset, expanding on the existing cohort.

### 2.4. Ethical Considerations

Prior to linking data between MSDS^3D^ and AOK PLUS, the physicians ascertain that the patient has comprehended the information and ZKN obtains a written IC form from each patient. All records identifying the subject are kept confidential and will not be made publicly available. Approval from the Ethics Committee at the Technical University of Dresden and the Ministry of Saxony (SGB § 75) was obtained to generate and maintain the database. No further ethical approval is required on a study-specific basis.

## 3. Database Description

### 3.1. Setting

The MSDS-AOK PLUS database provides regional coverage of MS patients visiting ZKN in Dresden and insured by AOK PLUS in Saxony and Thuringia (Table 1). Administrative claims data are provided retrospectively starting from 1st January 2014, with continuous updates with approximately a six-month data gap, whereas MSDS^3D^ documents patient data prospectively from initial MS diagnosis until the patient’s last visit at the center. To use the MSDS-AOK PLUS database for research purposes, ZKN and AOK PLUS provide approval after the review of a study protocol outlining objectives, methods, and planned analyses. Upon approval, IPAM e.V. will extract the data based on a data extraction manual, defining inclusion and exclusion criteria and all variables and format, and conduct the analyses. Indirect access to the database by external parties is not possible.

### 3.2. Data Contents

A detailed overview of the data parameters available in the MSDS-AOK PLUS database is presented in Table 2. Notably, some data parameters may be sourced from both AOK PLUS and MSDS^3D^. For research purposes, recommended use of data from each source will be based on the strengths of each dataset. Clinical information, such as MS subtype or date of first MS diagnosis is most reliable as documented in MSDS^3D^, which more accurately captures clinical parameters as a result of documentation by the treating MS center. In contrast, parameters such as healthcare resource utilization, prescription medications, or the broader patient comorbidity profile of the patient are captured more completely in AOK PLUS, due to their direct relevance for reimbursement purposes. In the case of prescriptions, while MSDS^3D^ captures medications prescribed by the physician at the center, whether the prescription was filled in the pharmacy is captured by AOK PLUS, serving as a better indication, although not full confirmation, of compliance to medications. Administrative information such as date of birth or sex is expected to be consistent between datasets due to the high-quality documentation systems in place. In the event of significant inconsistencies in such variables, there is a possibility to quality control data parameters with the patients at the center. In the event of remaining discrepancies or unresolved quality checks, it is recommended that the data parameter is transformed to missing or the patient is excluded from the analysis sample, conditional on the significance of the data point with respect to patient selection or outcomes. 

### 3.3. Purpose & Applications

RWD can provide sample populations for MS research to complement clinical trial evidence, which is often collected among narrowly defined and controlled patient populations [1,3]. In contrast to randomized controlled trials (RCTs), RWE can help inform risk stratification, outcome definitions, and guide treatment choices or sequences among heterogeneous groups of patients in clinical practice. Furthermore, RWE allows patient follow up in the long term, which is advantageous in evaluating the risks and benefits of therapies [1]. In addition to its contribution to understanding disease behavior and clinical effectiveness, RWE increasingly serves an important role in health economic decision making [2]. While there are various applications of RWD in MS, the evaluation of many research questions holistically has been a challenge due to variability in data parameters among different RWD sources. The MSDS-AOK PLUS linked database is a step towards closing this data gap, by providing a complete picture of the patient’s clinical state, capturing disease severity indicators documented regularly at MS related visits at a specialized neurology center, as well as all-cause resource use and diagnoses based on the patients’ continued encounter with the healthcare system (Table 2).

#### 3.3.1. The MSDS-AOK PLUS Mission

The MSDS-AOK PLUS database was generated for the purpose of expanding real-world evidence capabilities in MS, providing data to support various research objectives to ultimately guide clinical decision making. The database can be an asset to determine efficacy and safety in routine clinical care, understand treatment patterns and disease behavior as a step towards personalized medicine, and inform on the generalizability of RCT results. Detailed applications of the MSDS-AOK PLUS database are described in Table 3.

#### 3.3.2. Use Case: Baseline Characteristics and Resource Consumption of a Selected MSDS-AOK PLUS Patient Cohort

To highlight the data capture of the MSDS-AOK PLUS database, we present a descriptive use case of a selected cohort of patients. From the initial cohort of enrolled patients, we selected those with an existing diagnosis for MS, who were continuously insured for at least three years, starting from 30 June 2018 to 30 June 2021 (study period). Using 1 January 2019 as an arbitrary index date, we described patient baseline characteristics in the six-month period prior to index and select healthcare resource use, costs, and DMT use in the 2.5-year period from index until the end of the study period.

Overall, 186 patients (76.34% female) were selected (Table 4). At baseline, patients had a mean age of 49.14 years and a median time from the first MS diagnosis to the index of 12.07 years. The majority of patients (77.42%) had relapse-remitting MS (RRMS) at a timepoint closest to index, whereas 9.68% had secondary progressive MS (SPMS), with 7.53% of patients included in the sample missing information on MS subtype in the inclusion period. Closest to index, MS patients had a mean EDSS of 3.35, ranging from 1 to 8. One third of patients suffered from hypertension (33.33%) and depression or anxiety (32.80%) each at baseline.

From January 2019 to June 2021, a mean of 12.99 outpatient visits per patient year (PY) were observed, approximately half of which were MS-related (mean 6.69 visits per PY). In the follow-up period, 71 (38.71%) and 61 (32.80%) of patients had at least one all-cause hospitalization or MS-related hospitalization, with a mean 0.40 and 0.28 hospital admissions per PY, respectively. Related to their disease, mean MS-associated costs were highest for DMT use at €12,718.29 per PY, followed by €1,699.37 per PY for all-cause inpatient admissions, and €431.00 per PY for outpatient visits, independent of specialty (Table 5). The most commonly used DMTs in the follow-up period were ocrelizumab and fingolimod (18.28% each) (Figure 2). Moreover, a similar proportion of patients received glatiramer acetate and dimethyl fumarate, respectively (11.83% vs. 11.29%).

## 4. Discussion

### 4.1. Significance of MSDS-AOK PLUS

A strong evidence base is essential for understanding MS behavior and treatment in the real world. There are various clinical and patient MS registries available in Europe, such as the Big MS Data (BMSD) network combining the Danish, Italian, and Swedish MS registries with MSBase and the French Observatory of Multiple Sclerosis (OFSEP), as well as alternative national registries including the German and UK MS Registries [34,35,36]. A detailed overview of MS registries for observational research is maintained by the Multiple Sclerosis Data Alliance (MSDA) Catalogue [37]. Furthermore, MS patients can be analyzed for a subset of research questions from a number of national and regional administrative claims databases covering a wide population of patient profiles. However, none of these data sources alone can provide a full clinical picture of patients together with information on the overall comorbidity status, concomitant non-MS related therapies and general health care resource utilization and costs. While a few real-world studies in MS have made use of linkage between data sources, often to supplement clinical information and/or HCRU and costs, standalone linked databases of claims and EMRs of patients with MS remain scarce. This is especially true in Europe, where to the best of our knowledge there is no database linking claims to EMRs for MS patients, comparable to the MSDS-AOK PLUS database. In the UK, the Clinical Practice Research Datalink (CPRD) offers linkage opportunities for the general population, but in MS only references linkage of primary care data from GPs with hospital data, therefore lacking MS specific data parameters [38]. Moreover, as CPRD is a primary care-driven EMR database, a broad capture of data as seen in the AOK PLUS claims a dataset covering all healthcare settings (GP, specialties, inpatient) is not available without linkage to further sources. In the US, prior attempts to link claims data for MS patients to EMRs were hindered by high levels of missingness on the clinical indicators, such as the EDSS score, an issue that is not confronted in our data thanks to the high-quality assessment of the EDSS in the MSDS^3D^ system [11]. As such, with limited data sources available to address most research questions in MS, the MSDS-AOK PLUS linked database is a step towards improving the data gap and expanding the capabilities of real-world evidence in the field.

By bringing together the strengths of an MS patient management system, documenting disease specific characteristics and relevant PROs and in-depth comorbidity records, healthcare resource use, and costs documented via a regional claims database, the MSDS-AOK PLUS database will serve a crucial role in health economic outcome studies. The importance of both clinical and economic data in such studies can be highlighted by recent observational studies which have utilized the MS-HRS to quantify the economic impact of MS [28,39,40]. Research using the MS-HRS has shown a similar economic burden among men and women affected by MS, and an increased economic burden with relapses and disease severity [41]. Overall, these studies point to the importance of considering patient disease characteristics to appropriately assess costs in economic evaluations. While the MS-HRS adds a societal perspective considering resources which may not be relevant for reimbursement purposes, it is limited by reliable recall periods and patient discretion. As such, claims data supplemented by AOK PLUS in the linked MSDS-AOK PLUS database can be used to provide an accurate overview of resource consumption among total and subpopulations of MS patients across longer follow-up periods.

### 4.2. Limitations

Despite its advantages, the MSDS-AOK PLUS database is accompanied by several limitations. MS patients included in the database are selected based on their health insurance, AOK PLUS, based particularly in the regions of Saxony and Thuringia, potentially presenting selection and regional bias. In addition, all patients included in the database attend ZKN, which is a specialized center of excellence and therefore may raise questions with regards to the representativeness of the MSDS-AOK PLUS cohort compared to the wider MS population in Germany. To minimize any bias, all eligible patients were recruited without pre-screening at the time of their appearance to ZKN. Moreover, due to uniform healthcare regulations and standard practices across Germany, patients included are expected to be generally representative of the broader MS population. Furthermore, the combined dataset will consist of data derived from routine medical practice and, as such, exhibit a degree of missing data, coding error, or unstructured data with regards to outpatient recording. Despite this, claims data are considered to be valid and of high quality, [42,43,44] and various algorithms for data structuring and extraction in MSDS^3D^ are in place. In MSDS^3D^, data fields are predefined for the different areas of clinical routine and stored with reference values, e.g., for laboratory data or vital signs. Thus, the entered data is not only structured but also directly checked for plausibility, and error messages are displayed in case of incorrect entries. Edit checks ensure that missing data are avoided, and data entries are further checked for validity. This form of data entry and recording has been established in the field of clinical trials for several years but has additionally been expanded for documentation in the context of clinical routine.

### 4.3. Data Access and Future Perspectives

Future studies with the MSDS-AOK PLUS linked dataset must be approved by the data providers (AOK PLUS, ZKN) and carried out through the university-affiliated research institute IPAM e.V. To date, the database has been used to investigate the validation of a claims data-based proxy for EDSS, assessment of hospitalization costs by EDSS score in an alternative use case analysis, the investigation of MS subtype coding in claims data, and validating efficacy of propensity score methodologies for generating balanced comparator populations based on confounders observed in claims data (publications to follow) [45,46,47]. Additional validation purposes planned include the comparison of congruency between claims data and patient medical records or validation of outcome tools such as the MS-HRS questionnaire.

As of February 2023, the dataset captures 500 linked MS patients. Patient recruitment is ongoing in an effort to expand the current sample and set the stage for future RWE studies. Long term future goals include the expansion of the database into further regions covered by multiple insurance providers or additional specialized MS centers.

## 5. Conclusions

The MSDS-AOK PLUS linked database presents a unique opportunity in MS, expanding the current landscape and potentials of RWE. By combining the strengths of administrative claims databases and detailed clinical information provided in MS patient charts, MSDS-AOK PLUS offers a more complete picture of the patients’ history and clinical state. The database is actively growing, with the overall purpose of expanding the capabilities of MS research in the real-world.

## Figures and Tables

**Figure 1 jcm-12-01441-f001:**
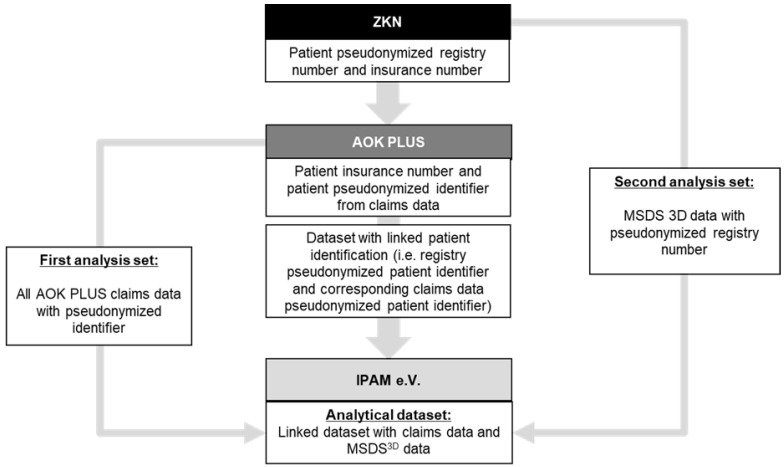
Data linkage process for the MSDS-AOK PLUS linked database.

**Figure 2 jcm-12-01441-f002:**
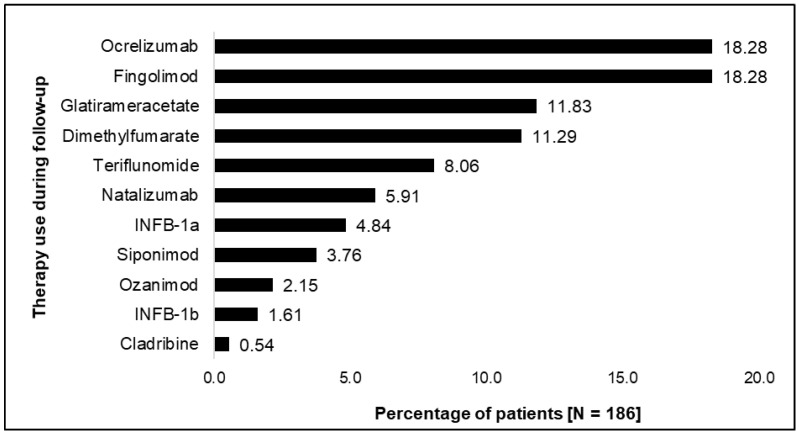
Disease modifying therapy use from index to the end of follow up of an example cohort in MSDS-AOK PLUS.

**Table 1 jcm-12-01441-t001:** Overview of MSDS-AOK PLUS coverage and access.

Parameter	Description
Coverage	Regional coverage of MS patients insured in the regions of Saxony and Thuringia visiting the ZKN center in Dresden
Data type	Retrospective claims and medical records supplemented with prospective data collection of medical records via MSDS^3D^
Sample size	500 patients ^a^
Data coverage	Administrative claims (AOK PLUS): 1 January 2014 to 31 March 2022 ^a^ Electronic Medical Records (MSDS^3D^): starting from initial MS diagnosis and onset of first symptoms, with regular documentation from patient’s first appearance at the clinic to date
Data access	Data available via access through university affiliated research institute (IPAM e.V.). Indirect access is not available
Ethical approval	Ethics Committee at the Technical University of Dresden and the Ministry of Saxonia (SGB § 75). No further ethical approval is required

^a^ As of February 2023; however, continuously updated.

**Table 2 jcm-12-01441-t002:** Detailed description of data parameters available in the MSDS-AOK PLUS linked database.

Data Field	Description	Data Source			
		AOK PLUS		MSDS^3D^	
		Availability	Details	Availability	Details
Sex	Categorical variable for male/female as documented in both databases	X		X	Documented at baseline
Date of birth	*AOK PLUS:* Year in which patient was born *MSDS^3D^:* Day, month, and year in which the patient was born	X		X	Documented at baseline
Weight	Weight in kg			X	Documented every 3–6 months
Height	Height in cm			X	Documented every 3–6 months
Date of initial MS diagnosis	Day, month, year of first MS diagnosis	Partial	If first diagnoses (ICD-10-GM) occur within data availability; based on proxy employing washout period	X	Documented at baseline; missing for some patients
Date of initial MS symptom	Day, month, year of onset of first MS symptom	Partial		X	Documented at baseline; missing for some patients
Type of first symptoms	*AOK PLUS:* Symptoms based on ICD-10-GM coding *MSDS^3D^:* Free text entry of type of symptoms observed at baseline	Partial	Based on available ICD-10-GM coding	X	Documented at baseline; missing for some patients
MS subtype	MS type coded as per ICD-10-GM as newly diagnosed/CIS, RRMS, PPMS, SPMS, or unspecified and development over time	X		X	Documented every 3–6 months if applicable
Concomitant diseases/diagnoses	*AOK PLUS:* Record of all diagnoses of patient under insurance based on inpatient and outpatient diagnoses *MSDS^3D^:* Record of MS symptoms and related comorbidities	X		Partial	Documented every 3–6 months if applicable
Death	Binary variable indicating whether the patient has died (yes/no); if yes, day, month, and year of death	X		X	
Lab tests	*AOK PLUS:* All laboratory tests documented through EBM codes *MSDS^3D^:* All laboratory tests that are run within patients’ visits at ZKN. Partial external lab data available if provided by patients.	X		Partial	Documented every 3–6 months depending on DMT use; conditional availability
Lab test results	All laboratory test results that are run within patients´ visits at ZKN. Partial external lab data available if provided by patients.			Partial	Documented every 3–6 months depending on DMT use; conditional availability
Imaging	*AOK PLUS:* All imaging procedures (MRI, OCT, etc.) as documented by OPS and EBM codes *MSDS^3D^:* MS specific imaging procedures including MRI and OCT	X		X	Documented every 3–6 months or once a year depending on DMT use
Imaging results	MS specific MRI and OCT images and neuroradiological findings over time as text describing amount and location of new lesions			X	Documented every 3–6 months or once a year depending on DMT use
General practitioner (GP) visits	*AOK PLUS:* GP visit documented by GP specialty code ^a^ *MSDS^3D^:* GP visits conditional on if physician letter is provided by patient	X		Partial	Conditional availability; diagnosis code not always available
Neurologist visits	*AOK PLUS:* Neurologists visit documented by Neurologist specialty code ^a^ *MSDS^3D^:* Neurologist visits taking place at ZKN. Partial documentation of external neurologist visits, depending on if physician letter is provided.	X		Partial	Documented every 3–6 months; conditional availability of external neurologist visits
Inpatient prescriptions	*AOK PLUS:* For expensive drugs, inpatient OPS or NUB codes are used to identify inpatient prescriptions of agents. *MSDS^3D^:* DMT use and other agents used to treat MS symptoms, conditional on if hospital discharge/physician letter is provided by patient	Partial		Partial	Documented every 3–6 months; conditional availability
Outpatient prescriptions	*AOK PLUS:* All prescriptions documented by ATC codes, with associated date of prescription, number, and strength of packages; treatment duration can be estimated. *MSDS^3D^:* DMT use and other agents used to treat MS symptoms	X		Partial	
Hospitalizations	*AOK PLUS:* All admissions and discharge dates of hospitalizations with associated diagnoses *MSDS^3D^:* Admission and discharge dates of hospitalization if information provided by the patient or with patient permission	X		Partial	Conditional availability
Emergency Room (ER) visits	*AOK PLUS:* ER visits admission and discharge dates associated with overnight inpatient admission (including diagnoses) or emergency services on evenings, weekends, or public holidays in the outpatient setting via EBM codes *MSDS^3D^:* Admission and discharge dates of ER visits if information provided by the patient or with patient permission	Partial		Partial	Conditional availability
Rehabilitation stays	*AOK PLUS:* All admission and discharge dates of inpatient rehabilitation stays with associated diagnoses paid by the sickness fund ^b^ *MSDS^3D^:* Admission and discharge dates of rehabilitation stays if provided by the patient or with patient permission	X		Partial	Conditional availability
Aids and remedies	*AOK PLUS:* Therapeutic devices (walkers, wheelchairs, etc.) captured via Hilfsmittel codes and remedy services provided by medically trained staff (physical, speech, occupational therapy, etc.) via Heilmittel codes *MSDS^3D^:* Partial capture of select aids (unstructured) ^c^	X		Partial	Aids every 3–6 months or yearly; remedies not currently available
Surgical procedures	*AOK PLUS:* Inpatient and outpatient procedures as documented by OPS and EBM codes, respectively *MSDS^3D^:* Any procedures or health-related assessments as provided by patients or with patient permission	X		Partial	Conditional availability
Non-surgical procedures	*AOK PLUS:* Inpatient and outpatient procedures as documented by OPS and EBM codes, respectively *MSDS^3D^:* Any procedures or health-related assessments as provided by patients or with patient permission	X		Partial	Conditional availability
Primary care costs (GP)	Total associated costs of GP visits documented by GP specialty code with associated diagnoses ^d^	X			
Outpatient specialist costs	Total associated costs of specialist visits (ex. neurologist) documented by physician specialty codes with associated diagnoses (all-cause vs. MS-specific) ^d^	X			
Inpatient admission costs	Costs reported per Diagnosis Related Group (DRG) codes, including total cost of inpatient stay with associated diagnoses (all-cause vs. MS-specific)	X			
Prescription costs	Cost associated to any outpatient prescriptions with associated ATC code	X			
Rehabilitation costs	Costs related to inpatient rehabilitation visits with associated diagnoses (all-cause vs. MS-specific) paid by the sickness fund ^b^	X			
Costs of aids & remedies	Costs of all therapeutic devices and remedy services by medically trained staff covered by the sickness fund	X			
Indirect costs due to sick leave	Based on number of days absent from work multiplied with average gross salary (all-cause vs. MS-specific)	X			
Relapses	Acute presentation of patients to ZKN at onset of new or worsening symptoms with subsequent evaluation to determine whether relapse or other cause is possible			X	General documentation every 3–6 months on disease activity
Expanded Disability Status Scale (EDSS) [5]	EDSS score (0–10) assessed by physicians with associated functional system sub-scores: Ambulation (0–10), Pyramidal (0–6), Cerebellar (0–5), Brainstem (0–5), Sensory (0–6), Bowel & Bladder (0–6), Visual (0–6), and Cerebral (0–5)			X	Documentation every 3–6 months; yearly for a subset of patients
MS Performance Test (MSPT) [27]	MSPT as a digital adaptation and expansion of the paper-based Multiple Sclerosis Functional Composite (MSFC)			X	Documentation every 3–6 months as of July 2017
	Processing Speed Test (PST), digital adaption to the Symbol Digit Modalities Test (SDMT)			X	Documentation every 3–6 months
	Low Contrast Letter Acuity Test (LCLAT), showing the number of correct letters at 100% and 2.5%			X	Documentation every 3–6 months
	Walking Speed Test (WST), digital adaptation to the Timed 25-Foot Walk (T25-FW); result as time in minutes			X	Documentation every 3–6 months
	Manual Dexterity Test (MDT), digital adaptation to the 9-Hole Peg Test (9HPT); result as time in minutes per hand			X	Documentation every 3–6 months
	Patient Determined Disease Steps (PDDS) [33], patient-reported outcome of disability in MS with nine ordinal levels ranging between 0 (normal) and 8 (bedridden)			X	Documentation every 3–6 months
	Quality of Life in Neurological Disorders (NeuroQoL) [29], patient-reported outcome on physical, mental, and social effects experienced by patients with MS (anxiety, depression, fatigue, upper extremity function—fine motor/activities of daily living, lower extremity function—mobility, applied cognition—general concerns and cognition–executive function, emotional and behavioral dyscontrol, positive affect and well-being; sleep disturbance, ability to participate in social roles and activities, satisfaction with social roles and activities, stigma).			X	Documentation every 6 months to yearly
MS-Health Resource Survey (HRS) [28]	Patient-reported healthcare resource utilization survey reporting direct medical costs, direct nonmedical costs) and indirect costs.			X	Documentation yearly
Early Mobility Impairment Questionnaire (EMIQ) [30]	Patient-reported questionnaire scores for nine-item EMIQ for detecting mobility impairment			X	Documentation yearly
Multiple Sclerosis Walking Scale (MSWS-12) [31]	Patient-reported MSWS-12 scores based on 12 item questionnaire to determine how much MS has limited walking ability in the past two weeks			X	Documentation yearly

^a^ Visits will be based on counted dates of invoiced EBM codes; ^b^ rehabilitation refers to inpatient rehabilitation stays paid by the sickness fund. Note that the majority of rehabilitation services are covered by the pension insurance; ^c^ documentation of all relevant aids and remedies in a structured format is planned as a future development, to be documented on each patient visit (every 3–6 months or once a year); ^d^ in the outpatient setting, services are invoiced using a system of weighted points. To assess the monetary payment, weighted points are multiplied by a uniform orientation value, defined by the National Association of Statutory Health Insurance Physicians; Additional abbreviations: ATC, Anatomical Therapeutic Chemical Classification; CIS, clinically isolated syndrome; DMT, disease-modifying therapy; EBM, German Uniform Evaluation Standard; EDSS, Expanded Disability Status Scale; ICD-10-GM, International Classification of Diseases, Tenth Revision—German Edition; MRI, magnetic resonance imaging; NUB, New Examination and Treatment Methods; OCT, optical coherence tomography; OPS, Operation and Procedure Classification System; PPMS, primary progressive multiple sclerosis; RRMS, relapsing-remitting multiple sclerosis; SPMS, secondary progressive multiple sclerosis.

**Table 3 jcm-12-01441-t003:** Applications of the MSDS-AOK PLUS database.

MSDS-AOK PLUS Applications
Comparative effectiveness
• Short/long-term effectiveness of therapeutic options or interventions • Effectiveness of therapeutic options or interventions among various subgroups or clinical indicators (by MS subtype, severity, or disability level (Expanded Disability Status Scale), relapses, treatment sequence) • Factors that determine or predict response/effectiveness
Post-authorization safety studies
• Safety and risk/benefit assessment • Comparative safety and tolerability between treatments or among treated vs. untreated patients • Factors that determine or predict safety/tolerability
Disease behavior & prognosis
• Identification of predictors of response or disability and investigation of disease severity (validation via ‘true’ patient outcomes in MSDS^3D^) • Assess effect of comorbidities and other confounders on disease behavior and prognosis
Treatment patterns
• Patient treatment history and treatment sequences, stratified by various subgroups or clinical indicators (by MS subtype, severity or disability level, relapses, prior treatments/interventions) to assess current landscape and address unmet needs
Economic modeling
• Healthcare resource utilization and costs, stratified by various subgroups or clinical indicators (by MS subtype, severity or disability level, relapses, treatment use) • Cost-effectiveness models
Outcome measure validation
• Generate definitions and proxies for outcome measures, severity indicators (Expanded Disability Status Scale) or diagnoses in isolated databases followed by validation using the linked dataset • Validation of outcome tools (ex. MS-Health Resource Survey patient survey) • Preparation of Pay-for-Performance Schemes by validation of respective outcome measures
Historical controls
• Replication of randomized controlled trials in the real-world setting (representativity) • Alternative standard of care comparators for single arm trials

**Table 4 jcm-12-01441-t004:** Baseline characteristics of an example cohort of MS patients in MSDS-AOK PLUS.

Characteristic	MS Cohort (N = 186)	Data Source
Female, n (%)	142 (76.34)	*AOK PLUS*
Age, years		*AOK PLUS*
Mean (SD)	49.14 (11.54)	
Median (Range)	50.03 (22.01–79.05)	
Time since first diagnosis ^a^, years		*MSDS^3D^*
Mean (SD)	12.07 (8.23)	
Median (Range)	11.01 (0.67–44.03)	
MS subtype ^b^, n (%)		*MSDS^3D^*
First manifestation/CIS	3 (1.61)	
RRMS	144 (77.42)	
PPMS	7 (3.76)	
SPMS	18 (9.68)	
Not specified	14 (7.53)	
Charlson Comorbidity Index (CCI) ^c^		*AOK PLUS*
0	117 (62.90)	
1–2	47 (25.26)	
3–4	16 (8.60)	
≥5	6 (3.23)	
EDSS		*MSDS^3D^*
Mean (SD)	3.35 (1.76)	
Median (Range)	3 (1–8)	
EDSS distribution, n (%)		*MSDS^3D^*
1–1.5	33 (17.74)	
2–2.5	51 (27.42)	
3–3.5	30 (16.1)	
4–4.5	22 (11.83)	
5–5.5	2 (1.08)	
6–6.5	26 (13.98)	
7–7.5	6 (3.23)	
≥8	1 (0.54)	
Not assessed	15 (8.06)	
Comorbidities at baseline, n (%)		*AOK PLUS*
Diabetes mellitus	18 (9.68)	
Cardiovascular disease	21 (11.29)	
Hypertension	62 (33.33)	
Obesity	18 (9.68)	
Depression & Anxiety	61 (32.80)	

^a^ If missing (n = 15), date of first symptom was used, otherwise excluded; ^b^ MS subtype recording closest to the index date; ^c^ components to calculate CCI can be found in Appendix A. Abbreviations: CIS, clinically isolated syndrome; EDSS, Expanded Disability Status Scale; MS, multiple sclerosis; PPMS, primary progressive multiple sclerosis; RRMS, relapsing-remitting multiple sclerosis; SD, standard deviation; SPMS, secondary progressive multiple sclerosis.

**Table 5 jcm-12-01441-t005:** Healthcare resource use and costs of an example cohort in MSDS-AOK PLUS from index to the end of follow up.

Outcome	MS Cohort (N = 186)
Follow-up time per patient (years)	2.50
Total patient years (PY)	464.24
**Outpatient Healthcare Resource Use**	
All outpatient visits	
Patients with ≥1 all-cause visit, n (%)	185 (99.46)
Mean all-cause visits per PY (SD)	12.99 (7.91)
Patients with ≥1 MS-related visit, n (%)	183 (98.39)
Mean MS-related visits per PY (SD)	6.69 (3.99)
GP visits	
Patients with ≥1 all-cause GP visit, n (%)	183 (98.39)
Mean all-cause GP visits per PY (SD)	3.49 (1.75)
Patients with ≥1 MS-related GP visit, n (%)	178 (95.70)
Mean MS-related GP visits per PY (SD)	3.12 (1.70)
Neurologist visits	
Patients with ≥1 all-cause visit, n (%)	51 (27.42)
Mean all-cause visits per PY (SD)	0.70 (1.38)
Patients with ≥1 MS-related visit, n (%)	44 (23.66)
Mean MS-related visits per PY (SD)	0.63 (1.32)
**Inpatient Healthcare Resource Use**	
Hospitalizations	
Patients with ≥1 all-cause admission, n (%)	72 (38.71)
Mean all-cause admissions per PY (SD)	0.40 (0.75)
Patients with ≥1 MS-related admission, n (%)	61 (32.80)
Mean MS-related admissions per PY (SD)	0.28 (0.57)
**Outpatient Costs, € per PY**	
All outpatient visits	
Mean cost of all-cause visits (SD)	1028.89 (798.17)
Mean cost of MS-related visits (SD)	431.00 (310.83)
GP visits	
Mean cost of all-cause GP visits (SD)	230.68 (116.04)
Mean cost of MS-related GP visits (SD)	206.28 (112.62)
Neurologist visits	
Mean cost of all-cause visits (SD)	44.71 (87.60)
Mean cost of MS-related visits (SD)	40.36 (84.05)
DMT prescriptions	
Mean cost of DMT prescriptions (SD)	12,718.29 (8933.00)
**Inpatient Costs, € per PY**	
Hospitalizations	
Mean cost of all-cause admissions (SD)	1699.37 (3871.59)
Mean cost of MS-related admissions (SD)	1507.24 (3676.63)

Abbreviations: DMT, disease modifying therapy; GP, general practitioner; MS, multiple sclerosis; PY, patient years; SD, standard deviation.

## Data Availability

Patient level data used for this study cannot be made publicly available in accordance with the local laws and policies of the participating institutions (AOK PLUS, ZKN, and IPAM e.V.).

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
