# Peer review of "Data Resource Profile: The Multiple Sclerosis Documentation System 3D and AOK PLUS Linked Database (MSDS-AOK PLUS)"

_jcm, 2023, doi:10.3390/jcm12041441_

Round 1
Reviewer 1 Report
This is really a novel approach to combine patient data in MS registries with data from health care providers and insurances which will be able to provide opportunities for better understanding clinically relevant descriptions of disease course and in the longer run better therapy options
Author Response
We would like to thank the reviewer for their positive evaluation and highlighting the impact of the MSDS-AOK PLUS dataset. We are pleased to read that you find it to be novel. Thank you very much again for your positive feedback.
Reviewer 2 Report
The author described and demonstrated how to use the data very well.
In the present case, the author show how to use the data link to search for specific questions in selected group of patients. I wonder if this data link canbe used as a predictor for treatment response and forecast disability for individual who present to the clinic. Could the author illustrate a case in those situations.
Author Response
Thank you to the reviewer for their positive feedback. The question posed regarding prediction of response and disability is an important point that we would like to further highlight in the manuscript. With true clinical information on response and disability available in MSDS3D for validation, the linked dataset can certainly be used to identify predictors that increase/decrease the probability of response or disability worsening. This has now been further clarified in Section 3.2, Table 3 under Purpose & Applications.
Reviewer 3 Report
Thank you for the interesting manuscript. It clearly describes a more complete database. The way in which this database is formed can also be interesting for other diseases.
I just have a few minor comments:
1/ For me it was not clear that Dresden is located in Saxony. I had to google it. Maybe this can be added for readers outside Germany.
2/ Figure 1: 'IPAM e.V.' Page 3 2.3 describes 'IPAM'. Is this the same since in figure 1 'e.V' is added?
3/ Table 2: Some data are available in both AOK Plus and MSDS. Theoretically, there could be a difference between both data (fore example because of wrong data entry or delay in data entry in 1 database) . How are differences handled?
4/ Page 3 of 17: rehabilitation costs: does this include physical therapy, speech therapy and occupational therapy from home?
5/ 3.2, line 176. 'RCT' is an abbreviation
Author Response
We would like to thank the reviewer for their positive feedback and noting the applicability of our approach to other disease areas.
With regards to the additional minor comments, many thanks for the outlined suggestions. We have implemented the input and believe the manuscript has improved as a result of the changes. We have now updated accordingly as follows:
1/ We have clarified that Dresden is located in Saxony in Section 2.1, describing MSDS3D.
2/ For consistency, we have updated all mentions of IPAM in the manuscript to “IPAM e.V.”
3/ This is a very important consideration, which we have now further elaborated under Section 3.1, describing Data Contents. Although some data may be available in both AOK and MSDS, data will be used from each dataset based on the strengths of the data source. Clinical information, such as MS subtype or date of first MS diagnosis, although theoretically can be derived from both datasets, the information is most reliable in MSDS3D, which better and more accurately captures clinical parameters as a result of documentation by the treating neurology center. In contrast, parameters such as healthcare resource utilization and costs, prescription medications, or the wider comorbidity profile of the patient (in all settings) are captured more completely in AOK PLUS, due to their direct relevance for reimbursement purposes. In the case of prescriptions, it is important to note that while in MSDS3D we are able to capture medications prescribed by the physician, from AOK PLUS we are able to detect whether the prescription was filled in the pharmacy (in addition to number and strength of packages) and serves as a better indication, although not confirmation of compliance. The information on prescriptions has been further updated in Table 2. Administrative information such as date of birth or sex is expected to be consistent between datasets due to the high-quality documentation systems in both datasets. In the event of significant inconsistencies in such variables, we would additionally have the possibility to quality control data parameters with the patients at the MS center. In the event of remaining discrepancies or unresolved quality checks, the data parameter will be transformed to missing or may require that the patient is excluded depending on the significance of the data point with respect to patient selection or outcomes.
4/ Rehabilitation stays and costs refer to inpatient rehabilitation only, which are paid by the sickness fund. Use of and respective costs of aids and remedies, which include physical therapy, speech therapy, and occupational therapy from home are captured separately under Hilfsmittel and Heilmittel codes, respectively in AOK PLUS, with only partial availability of aids in MSDS. These clarifications have been made in Table 2 (Section 3.1), including a separate item for aids and remedies.
5/ RCT has now been abbreviated at its first mention in Section 3.2
Thank you again for noting these important points.
Reviewer 4 Report
This study introduces a novel database which links administrative claims and medical records from an multiple sclerosis patient management system, allowing for complete capture of patient profiles. The dataset combines a complete record of patient diagnoses, treatment, healthcare resource use and costs (AOK PLUS) with detailed clinical parameters including functional performance and patient-reported outcomes (MSDS3D). The dataset currently captures 500 patients and hold the potential to increase the quality and scope of real-world studies in MS.
The manuscript is well-written and has a nice flow. The tables and illustrations are excellent. The methods are sufficiently described and the initial findings are clear and discussed straightforward. I have neither major or minor recommendations for the authors.
Author Response
We would like to express our gratitude to the reviewer for their evaluation of our manuscript. We are pleased to read that you have found the manuscript to be well-written and clear. Thank you again for the positive review.